# Recurrent episodes of pneumonia are associated with worse lung function in people with HIV

Sven J. Walderich[1*o], Jessica Fitzpatrick-Collins[2o], Aryana Bates[2‡],
Michelle H. Zhang[2‡], Jake Branchini[2‡], Kendall Gardner[2‡], Sharvari Bhide[2‡],
Amanda Jan[2‡], Rebecca Abelman[2,3o], Katerina L. Byanova[1o], Laurence Huang[1,2o]

1 Division of Pulmonary, Critical Care, Allergy and Sleep Medicine, Department of Medicine, University of California, San Francisco, California, United States of America, 2 Division of HIV, Infectious Diseases and Global Medicine, Department of Medicine, University of California, San Francisco, San Francisco, California, United States of America, 3 Medical Service, San Francisco Veteran Affairs Health Care System, San Francisco, California, United States of America

o These authors contributed equally to this work.
‡ AB, MHZ, JB, KG, SB, and AJ also contributed equally to this work.
* sven.walderich@ucsf.edu

## Abstract

### Background

HIV is associated with a decreased diffusing capacity for carbon monoxide (DLco), decreased spirometry, and an increased risk for developing obstructive lung disease, even in never-smokers. Pneumonia is a risk factor for impaired pulmonary function in people with HIV (PWH), but whether recurrent pneumonia results in a commensurate decline in lung function is unknown.

### Methods

We reviewed the electronic medical records and PFT data for PWH in San Francisco, CA. Participants were categorized by number of pneumonias since HIV diagnosis. We analyzed the association between the number of pneumonias and lung function using linear and logistic regression models after adjusting for substance use and HIV-associated factors.

### Results

Among 259 participants, the mean age was 51 years, 15% were female, and 42% were current smokers. One hundred (39%) participants had no prior pneumonia, 52 (20%) had 1 pneumonia, 50 (19%) had 2 pneumonias and 57 (22%) had ≥3 pneumonias. Multiple pneumonias were associated with a stepwise decline in FEV1, FVC, and DLCO %predicted (p-trend <0.001 for all) when controlling for demographic and clinical covariates. Having ≥3 pneumonias was associated with 2.84 higher odds of airflow obstruction (post-bronchodilator FEV1/FVC<LLN, 95% CI 1.02, 8.23) as well as 5.53 higher odds of impaired DLco (DLco<LLN, 95% CI 2.29, 14.0) compared to

**Data availability statement:** All relevant data are within the paper and its Supporting information files.

**Funding:** This work was supported by the National Institutes of Health (NIH) National Heart, Lung, and Blood Institute (NHLBI) [R01

HL128156, R01 HL128156-08S1, and R01 HL143998], all to Dr. Laurence Huang. Dr. Walderich is supported by T32 HL007185, Dr. Rebecca Abelman was supported by K12 HL143961, and Dr. Katerina Byanova is supported by F32 HL166065. The funders had no role in study design, data collection and analysis, decision to publish, or preparation of the manuscript.

**Competing interests:** The authors have declared that no competing interests exist.

those who never had pneumonia. There were no significant differences in PFT outcomes between participants with bacterial pneumonia versus *Pneumocystis* pneumonia during their first episode.

## Conclusion

In PWH with a high prevalence of inhalational substance use, recurrent episodes of pneumonia are associated with a commensurate decline in lung function in PWH characterized by airflow obstruction and diffusion limitations. While further studies are needed to elucidate the underlying mechanism, this work highlights the importance of preventing recurrent pneumonia in PWH in order to preserve lung function.

## Introduction

People with HIV (PWH) demonstrate higher rates of obstructive lung disease and impaired diffusing capacity for carbon monoxide (DLco), even when accounting for risk factors, including cigarette smoking [1–10]. Impaired DLco is the most common pulmonary function abnormality observed in PWH, a finding that persists despite virologic suppression on anti-retroviral therapy (ART) [5,6,8]. Among PWH who undergo both spirometry and DLco measurement, an isolated reduction in DLco with normal spirometry (iso↓DLco) is the most frequent pulmonary function pattern seen [11–13]. While the exact mechanisms that drive abnormal lung function in PWH are incompletely characterized, previously described risk factors include inhalational and intravenous substance use, impaired immune status, persistent immune activation as well as disease-related vascular complications such as pulmonary hypertension [2–4,8,9,14–18]. Furthermore, it has been shown that pulmonary infections may constitute a driver of abnormal lung function in PWH [19,20].

While opportunistic infections are less prevalent in the modern ART era, PWH remain at increased risk for bacterial pneumonia even in the setting of viral suppression [21,22]. In addition, limited healthcare access and medication non-adherence continue to place certain populations at risk for developing opportunistic pneumonias including *Pneumocystis* pneumonia (PCP) and tuberculosis (TB) [23,24]. Interestingly, several of the previously reported risk factors for abnormal lung function in PWH are also risk factors for recurrent and more severe pneumonia: advanced HIV infection, immune compromise, and substance use [10,22,25].

Prior studies have demonstrated that pneumonia may contribute to impaired pulmonary function in PWH. Morris *et al.* showed that PWH with a prior episode of bacterial pneumonia or PCP were at an increased risk of developing a persistent reduction in forced expiratory volume in one second (FEV1), forced vital capacity (FVC), FEV1/FVC, and DLco compared to PWH without prior pulmonary infections [19]. In contrast, a more recent study showed no permanent decline in lung function after an episode of PCP when controlling for smoking status, time since HIV diagnosis, and demographic factors [26].

Given this uncertainty, one of the objectives of this study was to examine the association between pneumonia and long-term pulmonary function outcomes among a prospective cohort of adults with HIV. We also sought to determine whether recurrent episodes of pneumonia result in a commensurate, additional decline in lung function and to characterize the pattern of pulmonary impairment following acute pneumonia in PWH. Specifically, we were interested in evaluating whether recurrent pneumonia is associated with an isolated diffusion limitation (iso↓DLco) independent of emphysema and obstructive lung disease. Finally, we investigated whether the type of pneumonia pathogen had an impact on the degree of pulmonary impairment.

## Methods

### Study cohort

Participants in this study were part of the Inflammation, Aging, Microbes, Obstructive Lung Disease and Diffusion Abnormalities (I AM OLD-DA) cohort, which is a longitudinal cohort study of PWH in San Francisco, California, USA and Kampala, Uganda evaluating change in lung function over time. This sub-study only involved participants from San Francisco. In San Francisco, the cohort is comprised of PWH ≥18 years of age who are cared for at Zuckerberg San Francisco General Hospital (ZSFG). Participants were recruited and enrolled during routine outpatient HIV clinic visits or when they were admitted to the hospital with acute pneumonia. Study visits occurred annually, and at each visit without evidence of acute pneumonia, participants underwent phlebotomy, standardized symptom assessments, and PFTs, consisting of pre- and post-bronchodilator spirometry and DLco measurement. For individuals enrolled at the time of acute pneumonia, baseline PFTs were performed three months after completion of treatment and only if acute symptoms had resolved. The Institutional Review Board of the University of California San Francisco approved the study protocol (IRB 13–11328) and participants provided their written informed consent.

### Data collection

We included all participants with spirometry efforts that met American Thoracic Society/European Respiratory Society (ATS/ERS) criteria for acceptability and reproducibility (grades A, B, or C) and at least one acceptable DLco measurement (grades A, B, C, or D) between April 2013 and December 2022 [27]. PFTs were performed by trained personnel in accordance with ATS/ERS standards and were overread by a single trained staff member. Spirometry measurements used in the analysis were obtained following bronchodilator administration (four 90 mcg actuations of albuterol sulfate). DLco measurements were adjusted for hemoglobin and carboxyhemoglobin measured at the time of PFTs. Percent predicted PFT values were obtained by adjusting measurements according to the GLI Global reference equations for spirometry and GLI reference equations for DLco [28]. If participants had multiple PFTs, we selected the first passing PFT.

Using the electronic medical record (EMR) at ZSFG, a single individual reviewed the charts of participants who met the study inclusion criteria using a predefined data collection tool between January 2023 and January 2024. The diagnosis of pneumonia (bacterial, *Pneumocystis*, tuberculous mycobacteria, non-tuberculous mycobacteria, fungal and viral including SARS-CoV-2) was classified as 'definitive' based on ATS/IDSA diagnostic guidelines or microbiological confirmation versus 'presumptive' based on compatible clinical presentation and illness resolution with empiric pathogen-specific therapy [29]. The date of each episode of pneumonia was documented to determine the relative timing compared to the participants' qualifying PFT as well as their HIV diagnosis. Only episodes of pneumonia occurring at or following the diagnosis of HIV were included. We also collected information regarding the type of pneumonia along with CD4 cell count, viral load, and ART adherence within the preceding 6 months prior to the diagnosis of pneumonia. For participants without an episode of pneumonia, ART use and viral load at the time of pulmonary function testing were used in the analyses. Additional data was collected from the EMR to assess for substance use as well as pre-existing lung disease, congestive heart failure, valvular heart disease, pulmonary hypertension, and diabetes with evidence of microangiopathy.

 

## Statistical analysis

Participants were grouped based on number of pneumonia episodes as having had zero, one, two, or three or more episodes of pneumonia. Demographic, clinical, and laboratory characteristics were summarized using counts with percent of the total (%) for categorical variables and medians with interquartile ranges (IQRs) for skewed continuous variables. We selected the following clinical predictors *a priori* based on literature review and known clinical associations: smoking history, inhalation or intravenous substance use, CD4 nadir as well as ART use and viral load at time of pneumonia.

We used univariable and multivariable linear regression models to estimate the association of number of episodes of pneumonia and FEV1 (in liters), FEV1%predicted, FVC (in liters), FVC %predicted, FEV1/FVC, DLco (in mL/min/mmHg) and DLco %predicted. The FEV1 and FVC measurements used in our analyses reflect post-bronchodilator efforts. The association of the number of pneumonia episodes and FEV1 < lower limit of normal (LLN), FVC < LLN, FEV1/FVC < LLN, and DLco < LLN was estimated using univariable and multivariable logistic regression models. We then performed sensitivity analyses to evaluate whether viral load at time of pneumonia or time between pneumonia and completion of PFTs were associated with a change in direction or magnitude of the observed association. Next, we analyzed the association between the type of pneumonia and FEV1%predicted, FVC %predicted, and DLco %predicted using linear regression models. Due to a reduced sample size, the analysis to evaluate the association between pneumonia type and pulmonary function outcomes was restricted to continuous outcomes.

We then investigated whether diffusion limitation is related to the development of obstructive lung disease or possibly reflects an independent lung function phenotype. For this, we used univariable and multivariable linear regression models to analyze the association between numbers of pneumonia and DLco among participants with normal spirometry (defined as FEV1/FVC ≥ LLN, FEV1 ≥ LLN and FVC ≥ LLN) and among those with airflow obstruction (defined as FEV1/FVC < LLN). Due to a small sample size, the analyses of those with abnormal spirometry should only be interpreted qualitatively for size and direction of effect.

## Results

### Study population

Among 332 eligible participants, 70 participants did not perform PFTs following an episode of pneumonia due to death, relocation, loss to follow-up, or other reasons (Fig 1). Three participants who underwent PFTs had post-bronchodilator

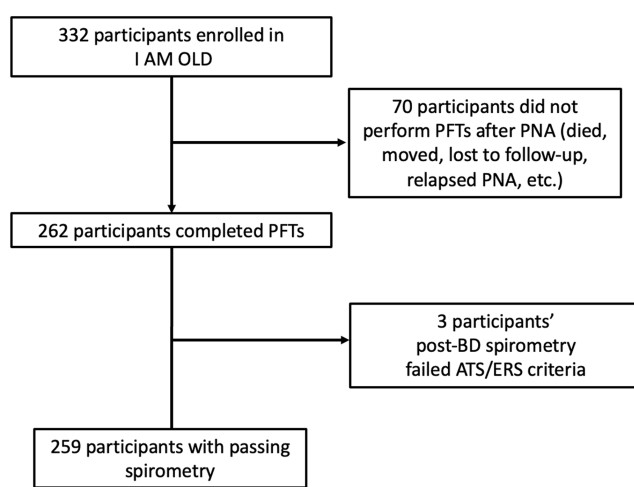

**Fig 1. Participant flow.**

spirometry that failed to meet ATS/ERS criteria of acceptability or reproducibility and were, therefore, excluded from the analysis. Ultimately, 259 participants had qualifying PFTs.

## Baseline demographics and clinical characteristics

Among the 259 included participants, the mean age was 51 years (Table 1). The majority were male (220, or 85%) and 39 (15%) were cisgender female. Eighty (31%) were former and 110 (42%) were current cigarette smokers at the time of PFT with an average smoking history of 19.5 pack-years. Two hundred and ten participants (81%) had used one or more inhalation drugs. Overall, the most commonly used inhaled substances aside from tobacco were marijuana (63%) followed by methamphetamine (42%), and cocaine (40%). One hundred and sixteen participants (45%) were current or former intravenous drug users. The mean duration from HIV diagnosis to the time of pulmonary function testing was 17 years (IQR 7, 26 years) and the median nadir CD4 count among the group was 161 cells/mm$^3$ (IQR 46, 345 cells/mm$^3$). At the time of the most recent episode of pneumonia, 55 participants (35%) had undetectable viral loads (<30 copies/mL) and 92 (58%) were consistently taking ART. At the time PFTs were obtained, 168 (65%) had undetectable viral loads and 187 participants (72%) were consistently taking ART.

A minority of patients had self-reported underlying pulmonary diseases including asthma (20%), chronic obstructive pulmonary disease (4%) and interstitial lung disease (1%). Fifteen participants (6%) had prior testing suggestive of pulmonary hypertension (defined as a mean pulmonary arterial pressure of >20 mmHg and a vascular resistance >2 Woods units during cardiac catheterization or suggested by an estimated right ventricular systolic pressure of >35 mmHg on echocardiography) and 65 patients (25%) had evidence of either systolic or diastolic left ventricular dysfunction based on a prior transthoracic echocardiogram. One hundred (39%) participants had no prior pneumonia, 52 (20%) had one, 50 (19%) had two, and 57 (22%) had 3 or more pneumonia episodes. The first pneumonia episode was caused by a typical bacterial organism in 89 (56%) participants, *Pneumocystis jirovecii* in 45 (28%) participants and a viral pathogen in 13 (8%) participants. Interestingly, the rate of pneumonia between participants who never used inhalational substances including tobacco and those with a history of prior or current use was similar (Table 2). The median time between the most recent episode of pneumonia and pulmonary function testing was 436 days (IQR 172, 1485 days).

## Association of number of pneumonia episodes and spirometry

Participants with a single episode of pneumonia had significantly lower post-bronchodilator FEV1%predicted and post-bronchodilator FVC %predicted compared to PWH who had no history of pneumonia (Table 3, Fig 2). The severity of impairment in FEV1 and FVC increased with each additional episode of pneumonia (p-trend <0.001) after adjustment for demographic and clinical covariates. FEV1%predicted in those with 3 or more prior episodes of pneumonia was lower by 24.2% (CI −16.2, −32.1%) compared to those with no prior episode. Similarly, having 3 or more prior episodes was associated with a reduction in FVC %predicted of 16.8% (CI −9.61, −24.0%) compared to participants with no prior pneumonia. Furthermore, participants with multiple prior episodes of pneumonia (≥3) were at 2.84 higher odds (95% CI 1.02, 8.23) of having airflow obstruction with a post-bronchodilator FEV1/FVC<LLN compared to participants with no prior pneumonia. When we isolated participants who had detectable viral loads at the time of the most recent pneumonia (N=91), the results appeared similar (Table 4). Furthermore, the direction and magnitude of the association between number of pneumonia episodes and PFT outcomes were not significantly different when controlling for the time between the last episode of pneumonia and the time at which PFTs were obtained.

## Association of number of pneumonia episodes and diffusing capacity

In addition to a significant impact on spirometry outcomes, our results demonstrate a clear association between prior episodes of pneumonia and a reduction in diffusing capacity. Compared to participants without prior pneumonia, the

**Table 1. Participant characteristics.**

| | N | Mean ± SD, Median (IQR) or N (%) |
|---|---|---|
| **Sociodemographic factors** | | |
| Age, years | 259 | 51 ± 12 |
| Female gender | 259 | 39 (15) |
| Race | 259 | |
| • Caucasian | | 102 (39) |
| • Black/African American | | 90 (35) |
| • Hispanic | | 43 (17) |
| • Asian | | 9 (3) |
| • Southeast Asian | | 5 (2) |
| • Other | | 10 (4) |
| Smoking History | 259 | |
| • Never | | 69 (27) |
| • Former | | 80 (31) |
| • Current | | 110 (42) |
| Pack-years | 190 | 19.5 ± 16.7 |
| Inhalation substances | 259 | |
| • None | | 49 (19) |
| • Cocaine | | 103 (40) |
| • Methamphetamines | | 108 (42) |
| • Opioids | | 7 (3) |
| • Marijuana | | 162 (63) |
| IV drug use | 259 | |
| • Never | | 143 (55) |
| • Current | | 39 (15) |
| • Former | | 77 (30) |
| **HIV factors** | | |
| Duration of HIV diagnosis (years) until PFT | 259 | 17 (7, 26) |
| Nadir CD4 count (cells/mm$^3$) | 259 | 161 (46, 345) |
| CD4 count at most recent PNA (cells/mm$^3$) | 151 | 340 ± 313 229 (101, 532) |
| CD4 count at time of PFT (cells/mm$^3$) | 259 | 505 ± 358 461 (246, 655) |
| Detected viral load (> 30 copies/mL) at time of most recent PNA | 159 | |
| • Detected | | 91 (57) |
| • Undetected | | 55 (35) |
| • Unknown | | 13 (8) |
| Detected viral load (> 30 copies/mL) at time of PFT | 259 | |
| • Detected | | 69 (22) |
| • Undetected | | 168 (65) |
| • Unknown | | 22 (8) |
| ART use (most recent PNA) | 159 | |
| • Yes | | 92 (58) |
| • No | | 57 (36) |
| • Unknown | | 10 (6) |

*(Continued)*

**Table 1.** (Continued)

| | N | Mean ± SD, Median (IQR) or N (%) |
|---|---|---|
| ART use (time of PFT) | 259 | |
| • Yes | | 187 (72) |
| • No | | 21 (8) |
| • Unknown | | 51 (20) |
| **Comorbidities** | | |
| Pulmonary Comorbidities | 259 | |
| • Asthma | | 52 (20) |
| • COPD | | 9 (4) |
| • Interstitial lung disease | | 3 (1) |
| Diabetes | 259 | |
| • Yes | | 16 (6) |
| • No | | 233 (90) |
| • Unknown | | 10 (4) |
| Pulmonary HTN | 259 | |
| • Yes | | 15 (6) |
| • No | | 92 (36) |
| • Unknown | | 152 (59) |
| LV dysfunction | 259 | |
| • Yes | | 65 (25) |
| • No | | 60 (23) |
| • Unknown | | 134 (52) |
| Valvular disease | 259 | |
| • Yes | | 13 (5) |
| • No | | 112 (43) |
| • Unknown | | 134 (52) |
| **PNA factors** | | |
| Number of episodes | 259 | |
| • 0 | | 100 (39) |
| • 1 | | 52 (20) |
| • 2 | | 50 (19) |
| • 3 or more | | 57 (22) |
| Type of PNA of the first episode | 159 | |
| • *Pneumocystis jirovecii* | | 45 (28) |
| • Typical bacterial | | 89 (56) |
| • Viral | | 13 (8) |
| • Other | | 12 (8) |
| Time from last PNA to PFT (days) | 159 | 436 (172, 1485) |

measured DLco (in mL/min/mmHg) and DLco %predicted in those with a single prior pneumonia episode was lower by 2.01 mL/min/mmHg (95% CI −0.14, −3.89 mL/min/mmHg) and 8.61%predicted (95% CI −1.44, −15.8%predicted), respectively, after adjustment for demographic and clinical covariates (Table 5, Fig 2). With increasing pneumonia episodes, we observed a progressive decline in DLco and DLco %predicted (p-trend <0.001). Multiple episodes of prior pneumonia

**Table 2. Incidence of pneumonia by inhalational substance use status.**

| Number of PNA | Never inhalational substance use (N = 49) | Current or past inhalational substance use (N = 210) | Never smoker (N = 69) | Current or past smoking (N= 190) |
|---|---|---|---|---|
| 0 | 17 (35%) | 83 (39%) | 24 (35%) | 76 (40%) |
| 1 | 10 (20%) | 42 (20%) | 16 (23%) | 36 (19%) |
| 2 | 11 (22%) | 39 (19%) | 16 (23%) | 34 (18%) |
| 3+ | 11 (22%) | 46 (22%) | 13 (19%) | 44 (23%) |

**Table 3. Association of number of PNA episodes and spirometry.**

| PNA episodes | Post-BD FEV1, L | Post-BD FEV1, % Pred | Post-BD FEV1 < LLN | Post-BD FVC, L | Post-BD FVC, % Pred | Post-BD FVC < LLN | Post-BD FEV1/FVC | Post-BD FEV1/FVC < LLN |
|---|---|---|---|---|---|---|---|---|
| **Unadjusted** | β (95% CI) | β (95% CI) | OR (95% CI) | β (95% CI) | β (95% CI) | OR (95% CI) | β (95% CI) | OR (95% CI) |
| 0 | Ref | Ref | Ref | Ref | Ref | Ref | Ref | Ref |
| 1 | **-0.46 (-0.78, -0.14)*** | **-7.40 (-14.2, -0.58)*** | 0.89 (0.30, 2.42) | **-0.57 (-0.94, -0.20)*** | **-7.69 (-13.7, -1.64)*** | 1.13 (0.28, 3.94) | 0.010 (-0.032, 0.043) | 0.76 (0.24, 2.00) |
| 2 | **-0.66 (-0.98, -0.34)*** | **-12.9 (-19.7, -6.09)*** | 1.63 (0.65, 4.04) | **-0.66 (-1.03, -0.29)*** | **-11.4 (-17.5, -5.38)*** | **2.85 (1.00, 8.49)*** | -0.024 (-0.061, 0.013) | 0.76 (0.24, 2.00) |
| 3+ | **-1.11 (-1.42, -0.80)*** | **-23.3 (-29.9, -16.7)*** | **4.56 (2.10, 10.3)*** | **-1.09 (-1.45, -0.73)*** | **-17.1 (-23.0, -11.3)*** | **3.63 (1.36, 10.4)*** | **-0.069 (-0.11, -0.033)*** | **2.27 (1.02, 5.10)*** |
| **Adjusted†** | aβ (95% CI) | aβ (95% CI) | aOR (95% CI) | aβ (95% CI) | aβ (95% CI) | aOR (95% CI) | aβ (95% CI) | aOR (95% CI) |
| 0 | Ref | Ref | Ref | Ref | Ref | Ref | Ref | Ref |
| 1 | **-0.31 (-0.57, -0.042)*** | **-9.38 (-17.4, -1.33)*** | 1.23 (0.40, 3.84) | -0.29 (-0.60, 0.01) | **-7.90 (-15.2, -0.60)*** | 1.26 (0.29, 4.83) | -0.010 (-0.048, 0.039) | 0.92 (0.28, 2.84) |
| 2 | **-0.39 (-0.66, -0.13)*** | **-14.3 (-22.2, -6.41)*** | 1.91 (0.67, 5.48) | **-0.35 (-0.65, -0.05)*** | **-11.2 (-18.4, -4.02)*** | **3.23 (1.00, 10.9)*** | -0.035 (-0.078, 0.010) | 0.88 (0.26, 2.84) |
| 3+ | **-0.65 (-0.92, -0.38)*** | **-24.2 (-32.1, -16.2)*** | **5.24 (2.00, 14.5)*** | **-0.52 (-0.83, -0.21)*** | **-16.8 (-24.0, -9.61)*** | **4.02 (1.31, 13.1)*** | **-0.079 (-0.12, -0.036)*** | **2.84 (1.02, 8.23)*** |
| *P-trend* | **<0.001** | **<0.001** | **<0.001** | **<0.001** | **<0.001** | 0.0029 | **<0.001** | 0.0035 |

Post-BD: post bronchodilator administration, LLN: lower limit of normal, FEV1: forced expiratory volume in one second, FVC: forced vital capacity.

†Adjusted for smoking history, inhalation or IV drug use, nadir CD4, ART use, viral load; measured FEV1 and FVC additionally adjusted for age, sex, and height. * P < 0.05.

were associated with significantly increased odds of having an abnormally low DLco (DLco<LLN) in unadjusted and fully adjusted models (2 or more episodes: aOR 2.68 95% CI 1.09, 6.74 and 3 or more episodes: aOR 5.53, 95% CI 2.29, 14.0). There was no significant change in the direction or magnitude of the association between number of prior pneumonias and diffusing capacity when the analysis was repeated among individuals with detectable viral loads at the time of their most recent pneumonia episode or when controlling for time between last episode of pneumonia and the time when PFTs were obtained (Table 6, Fig 3).

## Association of number of pneumonia episodes and diffusing capacity in participants with normal versus abnormal spirometry

Next, we wanted to evaluate whether the observed association between recurrent pneumonia and diffusion impairment is related to the development of obstructive lung disease or reflects an independent lung function phenotype, which we have termed iso↓DLco and defined as a DLco %predicted <LLN, FEV1/FVC>LLN, FEV1%predicted>LLN and FVC

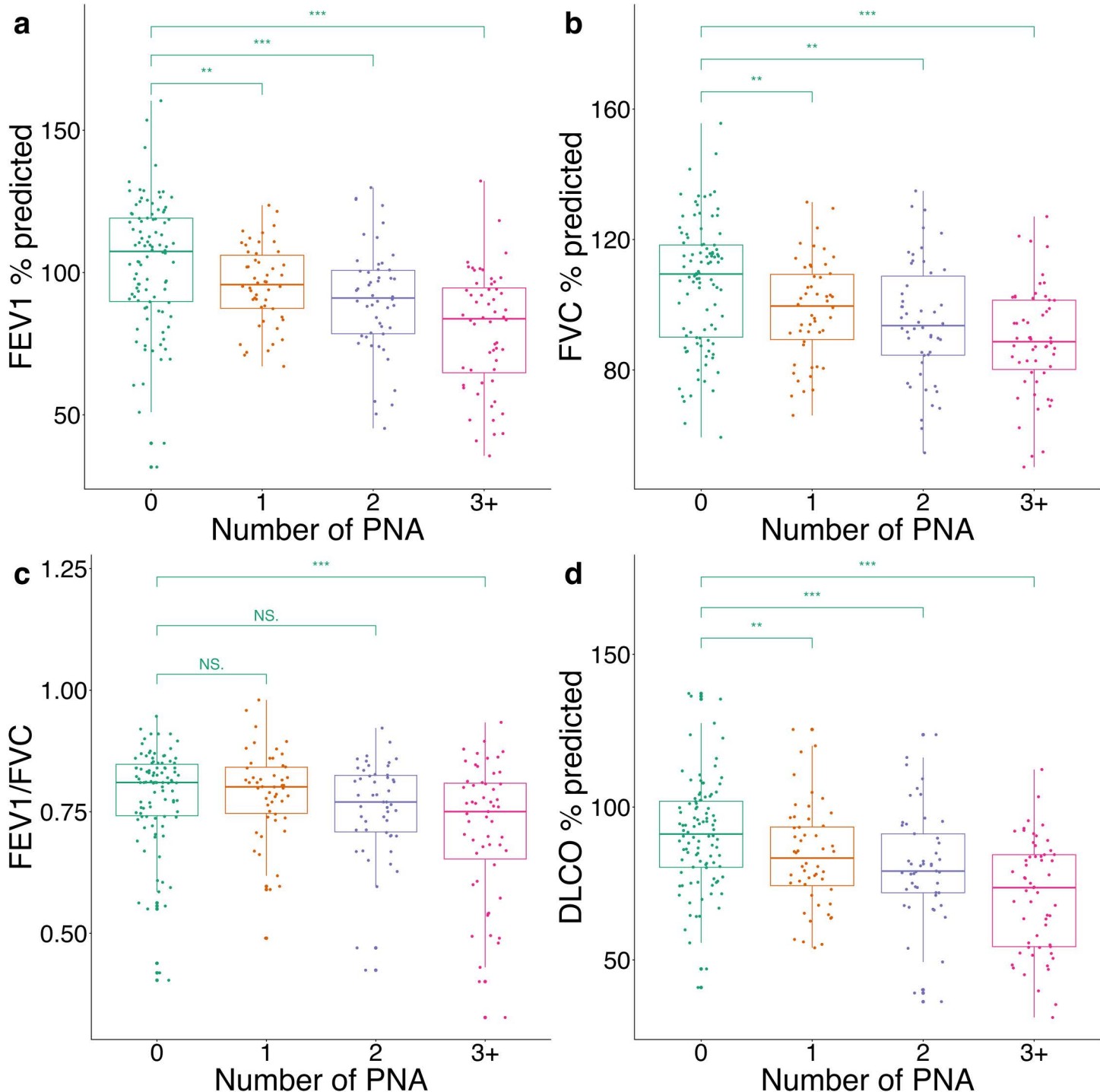

**Fig 2. Association of spirometry and DLco with recurrent episodes of PNA.** Distribution of percentage predicted FEV1 (**A**), percentage predicted FVC (**B**), FEV1/FVC (**C**), and percentage predicted DLCO (**D**) by number of PNA episodes. $P < 0.05$ (*); $P < 0.01$ (**); $P < 0.001$ (***).

**Table 4. Sensitivity analyses: association of number of PNA episodes and spirometry.**

| PNA episodes | Post-BD FEV1, L | Post-BD FEV1, % Pred | Post-BD FEV1 < LLN | Post-BD FVC, L | Post-BD FVC, % Pred | Post-BD FVC < LLN | Post-BD FEV1/ FVC | Post-BD FEV1/ FVC < LLN |
|---|---|---|---|---|---|---|---|---|
| **Sensitivity Analysis 1** | β (95% CI) | β (95% CI) | OR (95% CI) | β (95% CI) | β (95% CI) | OR (95% CI) | β (95% CI) | OR (95% CI) |
| 0 | Ref | Ref | Ref | Ref | Ref | did not converge | Ref | Ref |
| 1 | **-0.50 (-0.91, -0.10)*** | **-15.8 (-27.2, -4.39)*** | 3.12 (0.35, 70.2) | -0.38 (-0.83, 0.071) | **-12.4 (-22.8, -2.02)*** | | -0.043 (-0.11, 0.024) | 2.44 (0.36, 22.3) |
| 2 | -0.40 (-0.82, 0.032) | **-12.7 (-24.7, -0.80)*** | 1.34 (0.13, 31.9) | -0.25 (-0.72, 0.23) | **-11.2 (-21.9, -0.59)*** | | -0.040 (-0.11, 0.031) | 2.10 (0.28, 20.1) |
| 3+ | **-0.81 (-1.25, -0.36)*** | **-25.3 (-37.7, -12.8)*** | 4.76 (0.55, 10.8) | **-0.57 (-1.06, -0.076)*** | **-18.1 (-29.9, -7.08)*** | | **-0.096 (-0.17, -0.023)*** | **5.11 (0.77, 48.6)*** |
| *P-trend* | **<0.001** | **<0.001** | 0.07 | **0.045** | **0.011** | | **0.0015** | **0.019** |
| **Sensitivity Analysis 2** | | | | | | | | |
| 0 | Ref | Ref | Ref | Ref | Ref | Ref | Ref | Ref |
| 1 | **-0.30 (-0.57, -0.040)*** | **-8.79 (-16.8, -0.82)*** | 1.04 (0.28, 3.84) | -0.29 (-0.60, 0.015) | **-7.55 (-14.8, -0.26)*** | 0.98 (0.21, 3.97) | -0.014 (-0.056, 0.029) | 1.31 (0.38, 4.39) |
| 2 | **-0.39 (-0.65, -0.13)*** | **-11.5 (-19.3, -3.60)*** | 1.50 (0.48, 4.70) | **-0.34 (-0.64, -0.040)*** | **-10.8 (-18.1, -3.66)*** | 2.55 (0.74, 9.00) | -0.023 (-0.065, 0.019) | 0.97 (0.27, 3.29) |
| 3+ | **-0.63 (-0.90, -0.36)*** | **-20.0 (-28.1, -11.9)*** | **4.01 (1.36, 12.4)*** | **-0.49 (-0.80, -0.18)*** | **-15.7 (-22.9, -8.50)*** | 2.98 (0.90, 10.3) | **-0.065 (-0.11, 0.022)*** | **3.72 (1.27, 11.4)*** |
| *P-trend* | **<0.001** | **<0.001** | **0.0020** | **<0.001** | **<0.001** | **0.013** | **0.0034** | **0.0029** |

Sensitivity Analysis 1: Analysis carried out among those with detectable viral load at time of most recent PNA. Models adjusted for smoking history, inhalation or IV drug use, nadir CD4, ART use; measured FEV1 and FVC additionally adjusted for age, sex, and height.

Sensitivity Analysis 2: Analysis carried out among the overall group. Models adjusted for smoking history, inhalation or IV drug use, nadir CD4, ART use, viral load, and time between latest PNA and PFT; measured FEV1 and FVC additionally adjusted for age, sex, and height.

\* P < 0.05.

**Table 5. Association of number of PNA episodes and diffusion capacity.**

| PNA episodes | DLCO, mL/min/mmHg | DLCO % predicted | DLCO < LLN |
|---|---|---|---|
| **Unadjusted** | β (95% CI) | β (95% CI) | OR (95% CI) |
| 0 | Ref | Ref | Ref |
| 1 | **-3.24 (-5.27, -1.21)*** | **-7.10 (-13.1, -1.11)*** | 1.74 (0.81, 3.73) |
| 2 | **-3.98 (-6.05, -1.92)*** | **-10.6 (-16.6, -4.48)*** | **2.21 (1.04, 4.72)*** |
| 3+ | **-7.21 (-9.18, -5.24)*** | **-19.9 (-25.7, -14.1)*** | **4.54 (2.26, 9.37)*** |
| **Adjusted†** | aβ (95% CI) | aβ (95% CI) | aOR (95% CI) |
| 0 | Ref | Ref | Ref |
| 1 | **-2.01 (-3.89, -0.14)*** | **-8.61 (-15.8, -1.44)*** | 2.16 (0.90, 5.31) |
| 2 | **-2.04 (-3.93, -0.15)*** | **-9.96 (-17.1, -2.82)*** | **2.68 (1.09, 6.74)*** |
| 3+ | **-3.91 (-5.82, -2.00)*** | **-18.1 (-25.2, -11.1)*** | **5.53 (2.29, 14.0)*** |
| *P-trend* | **<0.001** | **<0.001** | **<0.001** |

†Adjusted for smoking history, inhalation or IV drug use, nadir CD4, ART use, viral load; measured DLCO additionally adjusted for age, sex, and height.

\*p<0.05.

%predicted>LLN (27–29). Among participants with normal spirometry, a single episode of pneumonia was associated with a lower DLco %predicted after adjustment for demographic and clinical covariates (aβ −6.45%predicted, 95% CI −13.2, 0.25%predicted) compared to individuals who had no prior pneumonia though this did not meet statistical significance

**Table 6. Sensitivity analyses: association of number of PNA episodes and diffusion capacity.**

| PNA episodes | DLCO, mL/min/mmHg | DLCO % predicted | DLCO < LLN |
|---|---|---|---|
| **Sensitivity Analysis 1** | β (95% CI) | β (95% CI) | OR (95% CI) |
| 0 | Ref | Ref | Ref |
| 1 | 1.76 (-5.01, 1.49) | -10.2 (-21.7, 1.35) | 2.26 (0.56, 10.5) |
| 2 | -1.68 (-5.12, 1.76) | -11.3 (-23.3, 0.66) | 2.40 (0.56, 11.6) |
| 3+ | **-4.40 (-7.92, -0.88)\*** | **-19.7 (-31.9, -7.49)\*** | 4.20 (0.99, 20.9) |
| *P-trend* | **0.0068** | **0.0031** | **0.079** |
| **Sensitivity Analysis 2** | | | |
| 0 | Ref | Ref | Ref |
| 1 | **-1.96 (-3.84, -0.074)\*** | **-8.39 (-15.5, -1.23)\*** | 2.28 (0.90, 5.83) |
| 2 | **-2.02 (-3.92, -0.13)\*** | **-9.89 (-17.0, -2.75)\*** | **2.51 (1.00, 6.41)\*** |
| 3+ | **-3.81 (-5.73, -1.88)\*** | **-17.7 (-24.9, -10.6)\*** | **4.92 (2.00, 12.7)\*** |
| *P-trend* | **<0.001** | **<0.001** | **0.0019** |

**Sensitivity Analysis 1:** Analysis carried out among those with detectable viral load at time of most recent PNA. Models adjusted for smoking history, inhalation or IV drug use, nadir CD4, ART use; measured DLCO additionally adjusted for age, sex, and height.

**Sensitivity Analysis 2:** Analysis carried out among the overall group. Models adjusted for smoking history, inhalation or IV drug use, nadir CD4, ART use, viral load, and time between latest PNA and PFT; measured DLCO additionally adjusted for age, sex, and height.

(Table 7). However, with each subsequent episode of pneumonia, we observed a significant, commensurate decrease in measured DLco (in mL/min/mmHg) and DLco %predicted (p-trend = 0.002 for DLco and p-trend <0.001 for DLco %predicted). Participants with normal spirometry and 3 or more episodes of pneumonia had significantly increased odds of having iso↓DLco with a DLco<LLN (aOR 4.99, 95% CI 1.38, 19.2) compared to those with normal spirometry but no prior pneumonia. A history of multiple episodes of pneumonia compared to no prior pneumonia was also associated with a trend (and similar magnitude) towards lower measured DLco and DLco %predicted among individuals with airflow obstruction (FEV1/FVC<LLN), however, these results did not reach statistical significance possibly due to the smaller sample size (Table 7).

### Impact of the type of pneumonia on lung function

We conducted a subgroup analysis among participants whose first episode of pneumonia was caused by either a typical bacterial organism (N = 89) or by *Pneumocystis jirovecii* (N = 45). Among these individuals, additional episodes of pneumonia beyond their first were associated with a commensurate decline in post-bronchodilator FEV1%predicted, post-bronchodilator FVC %predicted, and DLco %predicted consistent with the trend observed among the entire study population (Table 8). When comparing spirometry outcomes between individuals whose first pneumonia episode was caused by a typical bacterial pathogen to those caused by *Pneumocystis jirovecii*, we found no significant differences. While there was a trend towards lower DLco in individuals with PCP compared to those with typical bacterial pneumonia, this did not reach statistical significance.

### Discussion

In this prospective cohort of PWH with a high prevalence of inhalational substance use, we demonstrate that a single episode of pneumonia is associated with persistent impairment of lung function, including FEV1%predicted, FVC %predicted, and DLco %predicted. Furthermore, we also found that each subsequent episode of pneumonia is correlated with a commensurate, additional decline in pulmonary function. Our results suggest that recurrent episodes of pneumonia in PWH are associated with worsened lung function, and an increased risk of developing obstructive lung disease as well as

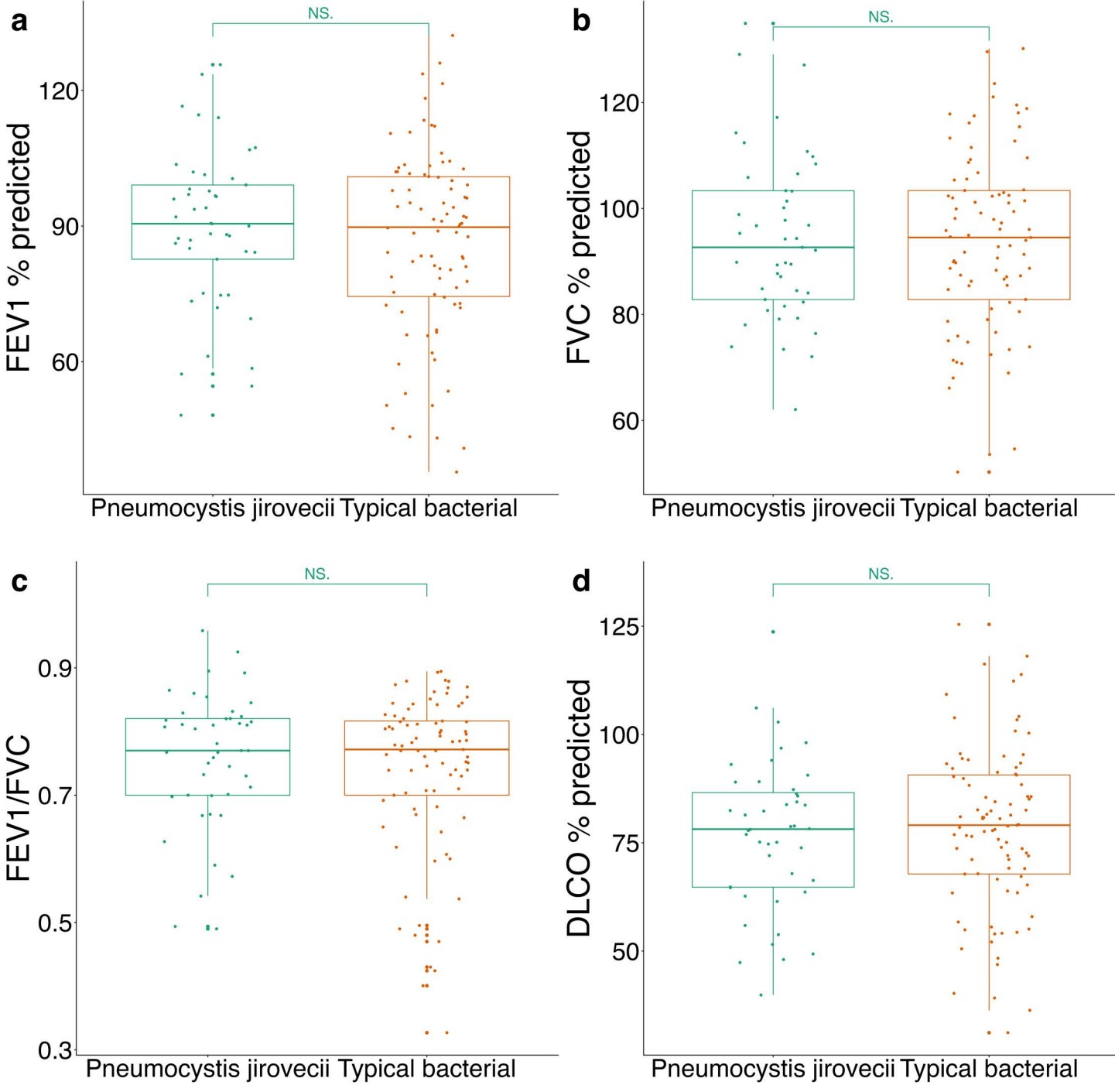

**Fig 3. Association of spirometry and DLco and type of first PNA.** Distribution of percentage predicted FEV1 (**A**), percentage predicted FVC (**B**), FEV1/FVC (**C**), and percentage predicted DLCO (**D**) by type of PNA of the first episode.

**Table 7. Association of number of PNA episodes and respiratory function among those with normal and abnormal spirometry.**

| PNA episodes | Normal Spirometry (FEV1/FVC ≥ LLN & FEV1 ≥ LLN & FVC ≥ LLN; N=181) | | | Abnormal Spirometry (FEV1/FVC < LLN; N=43) | | |
|---|---|---|---|---|---|---|
| | DLCO, mL/min/mmHg | DLCO % predicted | DLCO < LLN | DLCO, mL/min/mmHg | DLCO % predicted | DLCO < LLN |
| Unadjusted | β (95% CI) | β (95% CI) | OR (95% CI) | β (95% CI) | β (95% CI) | OR (95% CI) |
| 0 | Ref | Ref | Ref | Ref | Ref | Ref |
| 1 | **-2.41 (-4.55, -0.30)*** | -5.83 (-11.9, 0.21) | 1.38 (0.49, 3.68) | -3.08 (-9.40, 3.23) | -6.78 (-25.2, 11.6) | 3.00 (0.44, 27.1) |
| 2 | **-4.62 (-6.81, -2.42)*** | **-12.2 (-18.4, -6.02)*** | 2.05 (0.78, 5.35) | -3.10 (-9.85, 3.66) | -11.4 (-31.1, 8.26) | 6.00 (0.67, 134) |
| 3+ | **-6.23 (-8.57, -3.89)** | **-15.8 (-22.4, -9.16)*** | 2.67 (1.00, 7.14)* | **-6.27 (-11.0, -1.57)*** | **-16.7 (-30.4, -3.05)*** | 6.50 (1.39, 38.4)* |
| Adjusted† | aβ (95% CI) | aβ (95% CI) | aOR (95% CI) | aβ (95% CI) | aβ (95% CI) | Did not converge |
| 0 | Ref | Ref | Ref | Ref | Ref | |
| 1 | -1.64 (-3.43, 0.16) | -6.45 (-13.2, 0.25) | 1.99 (0.62, 6.53) | -1.08 (-6.56, 4.40) | -1.96 (-25.2, 21.3) | |
| 2 | **-2.99 (-4.98, -1.00)*** | **-12.1 (-19.4, -4.72)*** | 3.53 (1.03, 12.8)* | -0.99 (-6.41, 4.42) | -11.4 (-33.2, 10.5) | |
| 3+ | **-3.66 (-5.87, -1.45)*** | **-15.7 (-23.7, -7.64)*** | 4.99 (1.38, 19.2)* | -1.67 (-6.07, 2.74) | -14.3 (-31.6, 2.88) | |
| P-trend | **0.0016** | **<0.001** | **0.015** | 0.44 | 0.062 | |

†Adjusted for smoking history, inhalation or IV drug use, nadir CD4, ART use, viral load; measured DLCO additionally adjusted for age, sex, and height.

**Table 8. Association of PNA type and respiratory function.**

| PNA episodes | Post-BD FEV1 % predicted | Post-BD FVC % predicted | DLCO % predicted |
|---|---|---|---|
| | β (95% CI) | β (95% CI) | β (95% CI) |
| 1 | Ref | Ref | Ref |
| 2 | -3.86 (-12.0, 4.28) | -1.38 (-8.70, 5.95) | 1.76 (-6.09, 9.62) |
| 3+ | **-13.8 (-22.1, -5.51)*** | -6.36 (-13.8, 1.13) | -5.56 (-13.5, 2.35) |
| PNA type | | | |
| PJP | Ref | Ref | Ref |
| Typical bacterial | 0.92 (-7.02, 8.85) | 0.55 (-6.80, 7.89) | 4.53 (-3.32, 12.4) |

Subgroup analysis of those who had typical bacterial (N = 89) or pneumocystis (N = 45) pneumonia as their first episode.

†Adjusted for smoking history, inhalation or IV drug use, nadir CD4, ART use, viral load.

*p<0.05.

diffusion limitations. We found no significant difference in spirometry outcomes or diffusion capacity between participants who had typical bacterial pneumonia versus PCP as their initial pneumonia episode. Finally, while our results suggest that recurrent pneumonia increases the risk of obstructive lung disease in PWH, they also highlight that some develop a distinct phenotype of impaired lung function characterized by isolated diffusion limitation in the absence of airflow obstruction (iso↓DLco).

Previous work has shown conflicting data regarding the impact of pneumonia on long-term lung function in PWH. Morris *et al* revealed that a single prior episode of bacterial pneumonia or PCP was associated with an increased risk of persistent diffusion limitations, spirometric abnormalities, and airflow obstruction [19]. In contrast, a more recent study showed no permanent decline in lung function after an episode of PCP when controlling for smoking status, time since HIV diagnosis, and demographic factors [26]. Our study results are in agreement with the Morris study showing that even a single prior episode of pneumonia is associated with a statistically significant decline in post-BD FEV1%predicted, post-BD FVC %predicted, and DLco %predicted compared to PWH without prior pneumonia. Importantly, our findings build on the existing literature by showing that recurrent pulmonary infections in PWH are associated with a decline in

lung function that is proportionate to the number of prior pneumonia episodes. The precise reason for these apparently conflicting results is unclear, but a potential explanation may relate to our finding that there is no significant difference in the degree of pulmonary function impairment following bacterial pneumonia versus PCP – a finding that is consistent with the data presented by Morris *et al*. It is conceivable that varying numbers of prior bacterial pneumonia, which were not accounted for by Oomen *et al.*, could have limited the ability to detect differences in lung function related to a prior episode of PCP.

The mechanism by which recurrent pulmonary infections can lead to airflow obstruction or diffusion limitations remains uncertain. Studies have suggested that persistent airflow obstruction following pneumonia could be related to small airway dysfunction and airway hyperreactivity, which has been described both among people with advanced HIV in the pre-ART era and in people with well-controlled HIV infection [7,30–33]. In addition, recurrent pulmonary infections may exacerbate immune dysregulation and drive persistent inflammation [14]. It is possible that an aberrant immune response to recurrent pulmonary infections leads to excessive alveolar destruction. The loss of elastic recoil, in turn, could contribute to airflow obstruction due to the collapse of large airways, while a decrease in the alveolar-capillary interface would lead to a persistent decline in diffusing capacity. Both theories seem consistent with prior studies which showed that compared to never smoking people without HIV, never smoking PWH are more likely to have radiographic evidence of emphysema as well as small airway disease as evidenced by air trapping [7,34].

The finding that recurrent pneumonia can result in the development of an isolated reduction in the diffusing capacity without concomitant airflow obstruction carries important clinical implications as prior studies have shown that iso↓DLco is associated with increased morbidity and mortality [11,12,35,36]. Furthermore, it suggests the possibility that two distinct mechanisms exist that separately drive the development of either airflow obstruction or diffusion limitations in PWH following pulmonary infections. A prior study from our group revealed that distinct plasma biomarker profiles are associated with post-bronchodilator spirometry (FEV1%predicted) compared to diffusion capacity (DLco %predicted) in PWH as well as DLco %predicted when spirometry is normal compared to abnormal [13]. Interestingly, markers associated with an isolated reduction in DLco are largely related to macrophage activation and include soluble CD14 (a co-receptor for LPS), soluble CD163 (a cleavage product of metalloproteinases), TNF receptors 1 and 2 as well as IP10 (a marker of IFN response) [13,37–40]. Another study demonstrated that IFN signaling pathways are upregulated in PWH who have a reduced diffusing capacity [41]. In contrast, elevated interleukin-6 and its downstream acute phase reactant CRP were associated with a reduction in FEV1%predicted, a marker of airflow obstruction. IL-6 is a non-specific mediator of acute inflammation and has been shown to play an important role in the transition from acute to chronic inflammation, however, it is also a major product and activator of macrophages [42]. Given the association of both biomarker signatures with macrophage activation, abnormal macrophage function may play a central role in the pathogenesis of lung disease in PWH. This theory is even more intriguing considering that macrophages are functionally impaired even when people are virologically suppressed on ART and play a critical role in pneumonia as the first line of defense against pathogens [43,44]. Further studies are needed to investigate the potential role of aberrant macrophage function following pulmonary infections in PWH and to better understand how this results in distinct biomarker profiles as well as phenotypes of lung disease.

While pneumonia incidence in PWH has improved in the contemporary ART era, it remains a significant source of morbidity and mortality for this population. More broadly, pneumonia prevention through the routine use of vaccination remains an important public health priority. Pneumococcal vaccination is recommended for all adults with HIV due to its efficacy in pneumonia prevention. Unfortunately, we did not have access to vaccination records in this analysis and could not assess the role of vaccination on pneumonia prevalence, and, indirectly, lung function decline. However, it is conceivable that adequate vaccination against pneumococcus, SARS-CoV-2 or influenza may be even more critical in PWH than in people without HIV for the preservation of lung function.

A limitation of our study is the uncertainty surrounding the true duration of HIV infection in our participants, as individuals enrolled earlier were more likely to be diagnosed with HIV in the context of advanced disease and opportunistic

infections while participants enrolled later were more often diagnosed by routine HIV screening tests. An underestimation of the duration of HIV infection could result in missed episodes of pneumonia and thereby underestimate the effect size that pneumonia has on lung function abnormalities. Another limitation of this study is the high rate of smoking and inhalational substance use among our participants which affects the generalizability of our findings. While the self-reported prevalence of baseline chronic obstructive lung disease was low and the rate of pneumonia was similar with respect to smoking or inhalational substance use history, further studies are needed that include a larger proportion of non-smokers. Given the cross-sectional nature of this study, we were unable to establish causality and many of our measures relied on self-report. Lastly, selection bias may have occurred if, for example, participants with respiratory symptoms related to abnormal lung function were more likely to attend outpatient visits and be enrolled in our study, leading to an overestimation of the magnitude of our findings. A major strength of our study is that it involves a well-established, longitudinal cohort of patients recruited both from clinic and during hospitalization for acute pneumonia. Another strength is that lung function testing was conducted at a single site at predefined, standardized intervals following recovery from pneumonia and over-read by a single trained staff member, ensuring internal consistency and reliability.

In conclusion, we found that recurrent episodes of pneumonia in PWH with a high prevalence of inhalational substance use are associated with a commensurate and persistent decline in lung function. Multiple pulmonary infections are an independent risk factor for the development of obstructive lung disease and diffusion limitations in this population. The mechanism behind these findings remains incompletely understood. While further studies are needed to elucidate the underlying mechanism, this work clearly shows the importance of preventing recurrent pneumonia in PWH in order to preserve lung function.

## Supporting information

**S1 Dataset. PNA dataset.**
(XLSX)

## Acknowledgments

The authors would like to thank Jenny Hsieh and Eula Lewis who contributed to this work as respiratory therapists and oversaw pulmonary function testing throughout our study. We also thank Wayne Weeks who served as the pulmonary function testing technician and conducted all PFTs. Lastly, we would like to thank Maria Tercero Paz who worked with our group as a research coordinator.

## Author contributions

**Conceptualization:** Sven J. Walderich, Rebecca Abelman, Katerina L. Byanova, Laurence Huang.

**Data curation:** Sven J. Walderich, Aryana Bates, Michelle H. Zhang, Jake Branchini, Kendall Gardner.

**Formal analysis:** Jessica Fitzpatrick-Collins.

**Funding acquisition:** Laurence Huang.

**Investigation:** Sven J. Walderich.

**Methodology:** Rebecca Abelman, Katerina L. Byanova, Laurence Huang.

**Supervision:** Laurence Huang.

**Writing – original draft:** Sven J. Walderich.

**Writing – review & editing:** Jessica Fitzpatrick-Collins, Aryana Bates, Michelle H. Zhang, Jake Branchini, Kendall Gardner, Sharvari Bhide, Amanda Jan, Rebecca Abelman, Katerina L. Byanova, Laurence Huang.

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
