## [Decision Letter · Decision Letter 0]

27 Aug 2025

PONE-D-24-50359

Recurrent Episodes of Pneumonia Are Associated with Worse Lung Function in People with HIV

PLOS ONE

Dear Dr. Walderich,

Thank you for submitting your manuscript to PLOS ONE. After careful consideration, we feel that it has merit but does not fully meet PLOS ONE’s publication criteria as it currently stands. Therefore, we invite you to submit a revised version of the manuscript that addresses the points raised during the review process.

We look forward to receiving your revised manuscript.

Kind regards,

Yoshiaki Zaizen, MD, PhD

Academic Editor

PLOS ONE

Journal Requirements:

“This work was supported by the National Institutes of Health (NIH) National Heart, Lung, and Blood Institute (NHLBI) [R01 HL128156, R01 HL128156-08S1, and R01 HL143998], all to Dr. Laurence Huang. Dr. Walderich is supported by T32 HL007185; Dr. Rebecca Abelman was supported by K12 HL143961, and Dr. Katerina Byanova is supported by F32 HL166065.”

Additional Editor Comments:

The reviewers' comments will be valuable in improving your manuscript.

Please respond to the reviewers' comments by taking the necessary steps.

Reviewer's Responses to Questions

**Comments to the Author**

1. Is the manuscript technically sound, and do the data support the conclusions?

Reviewer #1: Partly

Reviewer #2: Partly

2. Has the statistical analysis been performed appropriately and rigorously?

Reviewer #1: Yes

Reviewer #2: Yes

3. Have the authors made all data underlying the findings in their manuscript fully available?

Reviewer #1: No

Reviewer #2: Yes

4. Is the manuscript presented in an intelligible fashion and written in standard English?

Reviewer #1: Yes

Reviewer #2: Yes

5. Review Comments to the Author

Reviewer #1: Overall a useful study on the natural history of pulmonary function in PWH with recurrent pneumonias. However I have a few major concerns reg conclusions and generalizability that can be addressed.

1. This cohort had a very high rate of current or past smoking (~70+%) and inhalation agent use. While this was 'adjusted' in the models, it is difficult to adjust for such a major common confounder. As such, I recommend conclusion of generalizability to all PWH (first paragraph under discussion and in abstract) should be toned down and caveat of "in this cohort of PWH with a very high prevalence of current/past smoking" should be mentioned. Also if numbers allow, a sensitivity analyses of excluding those with current/past smoking would be of interest.

2. Manuscript would have been much stronger if measures of QoL/physical functional state was also provided as an outcome (which the authors seem to have collected data on in their cohort). What is the clinical translation of, say, 10% reduction in predicted DLCO or FEV1?

3. given high prevalence of smoking, how was COPD handled? The lung function decline will depend heavily on background COPD severity and optimal management. At least provide data on COPD prevalence and adjust for.

4. Is there anyway to adjust/exclude viral pneumonias? Wonder if those would have a different effect?

5. Any data on vaccinations (viral or penumococcal) that can also affect the results and should be easy to extract from EMR?

Page 11: provide IQR after the sentence: "The median time between the most recent episode of pneumonia and pulmonary function testing was 148 days..."

Reviewer #2: Discussion

The discussion repeats many results rather than interpreting them. Please focus more on mechanisms, public health/clinical implications, and comparison with prior studies.

Limitations are acknowledged, but some key ones are missing (e.g., self-reported measures, cross-sectional design limiting causal inference, potential selection bias).

Conclusion

The conclusion should be more concise and aligned with the findings. Avoid making causal claims if the study design is observational.

6. PLOS authors have the option to publish the peer review history of their article (what does this mean?). If published, this will include your full peer review and any attached files.

Reviewer #1: No

Reviewer #2: **Yes:**Dr Dhiraj Agarwal

---

## [Author Response · Author response to Decision Letter 1]

17 Oct 2025

Reviewer 1

Comment 1: This cohort had a very high rate of current or past smoking (~70+%) and inhalation agent use. While this was 'adjusted' in the models, it is difficult to adjust for such a major common confounder. As such, I recommend conclusion of generalizability to all PWH (first paragraph under discussion and in abstract) should be toned down and caveat of "in this cohort of PWH with a very high prevalence of current/past smoking" should be mentioned. Also if numbers allow, a sensitivity analyses of excluding those with current/past smoking would be of interest.

Response : We agree with the feedback that our cohort had high rates of smoking and inhalational substance use which affects the generalizability of our findings and have revised our abstract as well as the first and final paragraph of our discussion accordingly. In reviewing our data, we did not find a significant difference in the incidence of pneumonia comparing participants who never used inhalational substances including tobacco with those who had a history of prior or current use. We have added these results as Table 1b. Unfortunately, our numbers were not sufficient to conduct a sensitivity analysis among never-smoking participants, though this would be an excellent cohort to investigate in a future study.

Revised Text: “In PWH with a high prevalence of inhalational substance use, recurrent episodes of pneumonia are associated with a commensurate decline in lung function in PWH characterized by airflow obstruction and diffusion limitations.” (page 3, lines 22-24)

“Interestingly, the rate of pneumonia between participants who never used inhalational substances including tobacco and those with a history of prior or current use was similar (Table 1b).” (page 11, lines 4-6)

“In this prospective cohort of PWH with a high prevalence of inhalational substance use, we demonstrate that a single episode of pneumonia is associated with persistent impairment of lung function, including FEV1 %predicted, FVC %predicted, and DLco %predicted. “ (page 14, lines 5-7)

“Another limitation of this study is the high rate of smoking and inhalational substance use among our participants which affects the generalizability of our findings. While the self-reported prevalence of baseline chronic obstructive lung disease was low and the rate of pneumonia was similar with respect to smoking or inhalational substance use history, further studies are needed that include a larger proportion of non-smokers.” (page 17, lines 19-21)

“In conclusion, we found that recurrent episodes of pneumonia in PWH with a high prevalence of inhalational substance use are associated with a commensurate and persistent decline in lung function.” (page 18, lines 10-12)

Comment 2: Manuscript would have been much stronger if measures of QoL/physical functional state was also provided as an outcome (which the authors seem to have collected data on in their cohort). What is the clinical translation of, say, 10% reduction in predicted DLCO or FEV1?

Response: This project had defined objectives focused on pneumonia/recurrent pneumonia/types of pneumonia (predictors) and lung function measurements (outcomes). Our group has previously shown that a reduced DLCO (predictor) is associated with increased respiratory symptom burden (modified Medical Research Council dyspnea scale, COPD Assessment Test score and St. George’s Respiratory Questionnaire) as well as increased mortality(outcomes) (PMIDs: 37463173 and 40504822), which are currently referenced in our discussion section. In addition, other studies have demonstrated an association between respiratory symptoms and a decline in spirometry (PMID 12762353), which is why we focused this study on objective lung function parameters. Your question is an important one, but also one that would require a different design (possibly longitudinal), as well as different predictors (change/reduction in lung function), and different outcomes (QoL). As we continue to gather more participants with more longitudinal values we plan to address the impact on quality of life measures in future projects.

Revised Text: “The finding that recurrent pneumonia can result in the development of an isolated reduction in the diffusing capacity without concomitant airflow obstruction carries important clinical implications as prior studies have shown that iso↓DLco is associated with increased morbidity and mortality (27,28,35,36)”. (page 16, lines 2-5)

Comment 3: Given high prevalence of smoking, how was COPD handled? The lung function decline will depend heavily on background COPD severity and optimal management. At least provide data on COPD prevalence and adjust for.

Response: We appreciate this feedback and have included data on baseline prevalence of self-reported COPD (n=9, 4%) as well as other pulmonary comorbidities in Table 1. It is true that the decline in lung function is affected by COPD. However, given the low baseline prevalence of COPD in our cohort we did not adjust for this comorbidity in our models but rather included smoking as a covariable. We did complete an analysis among participants without airflow obstruction and observed a similar association between recurrent pneumonias and a decline of DLCO (Table 4).

Revised Text: “A minority of patients had self-reported underlying pulmonary diseases including asthma (20%), chronic obstructive pulmonary disease (4%) and interstitial lung disease (1%).” (page 10, lines 19-20)

Comment 4: Is there anyway to adjust/exclude viral pneumonias? Wonder if those would have a different effect?

Response: This is an especially interesting point considering the recent COVID pandemic. Unfortunately, the number of viral pneumonias (13, or 8%) was insufficient to determine a differential impact on long-term lung function compared to other etiologies. We have added the prevalence of viral pneumonia to Table 1.

Comment 5: Any data on vaccinations (viral or pneumococcal) that can also affect the results and should be easy to extract from EMR?

Response: We thank the reviewer for this important point. We did not have immunization data for our study participants. We have added some additional text about the potential role of immunization on our findings in the discussion.

Revised Text: “While pneumonia incidence in PWH has improved in the contemporary ART era, it remains a significant source of morbidity and mortality for this population. More broadly, pneumonia prevention through the routine use of vaccination remains an important public health priority. Pneumococcal vaccination is recommended for all adults with HIV due to its efficacy in pneumonia prevention. Unfortunately, we did not have access to vaccination records in this analysis and could not assess the role of vaccination on pneumonia prevalence, and, indirectly, lung function decline. However, it is conceivable that adequate vaccination against pneumococcus, SARS-CoV-2 or influenza may be even more critical in PWH than in people without HIV for the preservation of lung function. “ (page 17, lines 4-12)

Comment 6: Provide IQR after the sentence: "The median time between the most recent episode of pneumonia and pulmonary function testing was 148 days..."

Response: We have included the IQR and added the data point to Table 1a. After reviewing our data, we discovered that the previously reported median time between the most recent episode of pneumonia and pulmonary function test (148 days) was inaccurate; this has now been corrected.

Revised Text: “The median time between the most recent episode of pneumonia and pulmonary function testing was 436 days (IQR 172, 1485 days).” (page 11, lines 6-8)

Reviewer 2

Comment 1: The discussion repeats many results rather than interpreting them. Please focus more on mechanisms, public health/clinical implications, and comparison with prior studies.

Response: We have revised the discussion by limiting the reiteration of results to the first paragraph and have added additional interpretation in the subsequent paragraphs. We have also added a paragraph on the public health and clinical implications.

Revised Text: “While pneumonia incidence in PWH has improved in the contemporary ART era, it remains a significant source of morbidity and mortality for this population. More broadly, pneumonia prevention through the routine use of vaccination remains an important public health priority. Pneumococcal vaccination is recommended for all adults with HIV due to its efficacy in pneumonia prevention. Unfortunately, we did not have access to vaccination records in this analysis and could not assess the role of vaccination on pneumonia prevalence, and, indirectly, lung function decline. However, it is conceivable that adequate vaccination against pneumococcus, SARS-CoV-2 or influenza may be even more critical in PWH than in people without HIV for the preservation of lung function. “ (page 17, lines 4-12)

Comment 2: Limitations are acknowledged, but some key ones are missing (e.g., self-reported measures, cross-sectional design limiting causal inference, potential selection bias).

Response: We have added additional limitations to the discussion section.

Revised Text: “Another limitation of this study is the high rate of smoking and inhalational substance use among our participants which affects the generalizability of our findings. While the self-reported prevalence of baseline chronic obstructive lung disease was low and the rate of pneumonia was similar with respect to smoking or inhalational substance use history, further studies are needed that include a larger proportion of non-smokers. Given the cross-sectional nature of this study, we were unable to establish causality and many of our measures relied on self-report. Lastly, selection bias may have occurred if participants with respiratory symptoms related to abnormal lung function were more likely to attend outpatient visits and be enrolled in our study, leading to an overestimation of the magnitude of our findings” (page 17, lines 19-25 and page 18, lines 1-3)

Comment 3: The conclusion should be more concise and aligned with the findings. Avoid making causal claims if the study design is observational.

Response: We appreciate this comment. We have revised our wording in the conclusion to avoid any claims of causality.

Revised Text: “In conclusion, we found that recurrent episodes of pneumonia in PWH with a high prevalence of inhalational substance use are associated with a commensurate and persistent decline in lung function.” (page 18, lines 10-12)

---

## [Decision Letter · Decision Letter 1]

24 Feb 2026

Recurrent episodes of pneumonia are associated with worse lung function in people with HIV

PONE-D-24-50359R1

Dear Dr. Walderich,

We’re pleased to inform you that your manuscript has been judged scientifically suitable for publication and will be formally accepted for publication once it meets all outstanding technical requirements.

Kind regards,

Yoshiaki Zaizen, MD, PhD

Academic Editor

PLOS One

Additional Editor Comments (optional):

Thank you for submitting your manuscript to PLOS One.

Reviewers' comments:

Reviewer's Responses to Questions

**Comments to the Author**

1. If the authors have adequately addressed your comments raised in a previous round of review and you feel that this manuscript is now acceptable for publication, you may indicate that here to bypass the “Comments to the Author” section, enter your conflict of interest statement in the “Confidential to Editor” section, and submit your "Accept" recommendation.

Reviewer #1: All comments have been addressed

2. Is the manuscript technically sound, and do the data support the conclusions?

Reviewer #1: Yes

3. Has the statistical analysis been performed appropriately and rigorously?

Reviewer #1: Yes

4. Have the authors made all data underlying the findings in their manuscript fully available?

Reviewer #1: Yes

5. Is the manuscript presented in an intelligible fashion and written in standard English?

Reviewer #1: Yes

6. Review Comments to the Author

Reviewer #1: here to bypass the “Comments to the Author” section,. all comments have been addressed. I have no competing interest to report

7. PLOS authors have the option to publish the peer review history of their article (what does this mean?). If published, this will include your full peer review and any attached files.

Reviewer #1: No

---

## [Editor Report · Acceptance letter]

PONE-D-24-50359R1

PLOS One

Dear Dr. Walderich,

I'm pleased to inform you that your manuscript has been deemed suitable for publication in PLOS One. Congratulations! Your manuscript is now being handed over to our production team.

Kind regards,

on behalf of

Dr. Yoshiaki Zaizen

Academic Editor

PLOS One